# Palmitic Acid Impairs Myogenesis and Alters Temporal Expression of miR-133a and miR-206 in C2C12 Myoblasts

**DOI:** 10.3390/ijms22052748

**Published:** 2021-03-09

**Authors:** Ailma O. da Paixão, Anaysa Paola Bolin, João G. Silvestre, Alice Cristina Rodrigues

**Affiliations:** 1Department of Pharmacology, Instituto de Ciencias Biomedicas, Universidade de Sao Paulo, Sao Paulo 05508-000, Brazil; ailmaoliveira@usp.br (A.O.d.P.); anaysa.bolin@usp.br (A.P.B.); 2Department of Anatomy, Instituto de Ciencias Biomedicas, Universidade de Sao Paulo, Sao Paulo 05508-000, Brazil; jgsilvestre@alumni.usp.br

**Keywords:** obesity, atrophy, fatty acids, skeletal muscle, microRNAs

## Abstract

Palmitic acid (PA), a saturated fatty acid enriched in high-fat diet, has been implicated in the development of sarcopenic obesity. Herein, we chose two non-cytotoxic concentrations to better understand how excess PA could impact myotube formation or diameter without inducing cell death. Forty-eight hours of 100 µM PA induced a reduction of myotube diameter and increased the number of type I fibers, which was associated with increased miR-206 expression. Next, C2C12 myotube growth in the presence of PA was evaluated. Compared to control cells, 150 µM PA reduces myoblast proliferation and the expression of *MyoD* and miR-206 and miR-133a expression, leading to a reduced number and diameter of myotubes. PA (100 µM), despite not affecting proliferation, impairs myotube formation by reducing the expression of Myf5 and miR-206 and decreasing protein synthesis. Interestingly, 100 and 150 µM PA-treated myotubes had a higher number of type II fibers than control cells. In conclusion, PA affects negatively myotube diameter, fusion, and metabolism, which may be related to myomiRs. By providing new insights into the mechanisms by which PA affects negatively skeletal muscle, our data may help in the discovery of new targets to treat sarcopenic obesity.

## 1. Introduction

A high-lipid diet, mainly containing saturated fatty acids, such as palmitic acid (PA), the second most abundant fatty acid in circulation, is associated with the development of obesity [1,2]. Excess delivery of PA to skeletal muscle is implicated in the development of insulin resistance resulting from lipid oversupply to skeletal muscle [3,4,5], inhibitory effects on the insulin signaling, mediated by Protein kinase C theta-dependent activation of mammalian target of rapamycin (mTOR)/p70 ribosomal S6 kinase (S6K) pathway [6] and loss of muscle mass [7,8,9]. Increased amounts of adipose tissue (AT) and the presence of sarcopenia, which is the loss of muscle mass and strength or physical function is a condition known as sarcopenic obesity. Sarcopenic obesity may have a greater effect on metabolic disorders and mortality than either sarcopenia or obesity alone.

A healthy skeletal muscle mass is essential in the reduction of complications of obesity. Lipid overload in muscle appears to affect not only insulin signaling but also muscle mass and muscle regenerative capacity. Previously, we have shown myogenesis is reduced and myomiRs are dysregulated in mouse soleus muscle after 12 weeks of high-fat diet [10], suggesting that microRNAs are associated with skeletal muscle loss in diabetic and obese subjects.

MicroRNAs are small non-coding RNAs that act on the post-transcriptional regulation of gene expression. The importance of microRNAs in metabolic diseases are well demonstrated by mice specifically lacking dicer in adipose tissue: these mice exhibit a defect in miRNA processing in AT, resulting in a decrease in white AT mass, whitening of brown AT mass, insulin resistance, and dyslipidemia [11]. By the same token, microRNAs are increasingly being characterized as important regulators in muscle formation, myogenic differentiation, skeletal muscle adaptation, and are already indicated as important biomarkers for muscular diseases [12]. However, which microRNAs can contribute to sarcopenic obesity is unknown.

Taking into consideration that chronic exposure to saturated fatty acids is deleterious to skeletal muscle, we chose two-non cytotoxic PA concentrations to better understand how excess PA could impact myotube formation or skeletal muscle mass by evaluating the expression of proteins and microRNAs involved in myogenesis.

Herein, we show that PA reduced myotube diameter in 5-day differentiated C2C12 cells, increased the number of type I fibers, and upregulated miR-206 expression. There was no difference in protein synthesis or levels of proteins involved in protein degradation. When C2C12 myoblasts were cultivated in differentiation media containing PA (100–150 µM), both concentrations resulted in smaller myotubes, but only a higher concentration of PA decreased myoblast proliferation. Of note, myotubes formed from PA-treated myoblasts were more glycolytic and had decreased levels of miR-206, as previously described in skeletal muscle of diabetic subjects.

## 2. Results

### 2.1. Effect of PA on Viability, DNA Fragmentation and Proliferation of C2C12 Cells

Initially, MTT (3-(4,5-dimethyl-2-thiazolyl)-2,5-diphenyl-2H-tetrazolium bromide) and lactate dehydrogenase (LDH) assays were performed with the purpose of evaluating cell viability of myoblasts treated with PA and vehicle, and to define the concentration that was not toxic for the cells. We first attempted to use 250 μM and 500 μM concentrations; however, these concentrations significantly reduced cell viability after 24 h (13% and 46%, respectively). Next, we did a time-course using lower concentrations of PA (50–150 μM). From analyses of both MTT and LDH assays (Figure 1A,B), we observed that PA treatment for 72 h from 50 to 150 μM does not affect cell viability. In order to evaluate if PA induced DNA fragmentation, a late phase of apoptotic cells, we performed a TUNEL (TdT-mediated dUTP-TMR Nick End Labeling) assay. We observed that there was no labeling of apoptotic cells in any of the groups evaluated (Appendix A).

Wound-healing assay was used to evaluate PA exposure on C2C12 cell proliferation. The scratch made resulted in a gap with similar distances (Control: 274.29 ± 52.66 µm; 100 µM PA: 274.63 ± 53.07 µm and 150 µM PA: 263.02 ± 65.75 µm, *p* > 0.05) (Figure 1C,D). In control and 100 µM PA groups, there were no differences along the timeline in cellular growth (Figure 1C,D). However, 150 µM PA reduced cell proliferation; at 12 h and 16 h, the gap sizes in PA-treated cells were 69% and 41%, respectively, while in vehicle-treated cells, the gap sizes were 18% and 3%, respectively (Figure 1C,D).

Altogether, these data suggest that PA (100 and 150 μM) does not cause cytotoxicity; however, cells treated with 150 μM PA showed a decrease in proliferation compared to control or 100 μM PA. Thus, we set 100 μM concentration to treat differentiated myotubes to avoid death and keep their proliferative capacity similar to the control myotubes.

### 2.2. PA Induces Smaller C2C12 Myotubes and Metabolic Remodeling

The effect of PA in the two main fiber types (I and II) was evaluated by immunolabeling of differentiated myotubes with anti-Myosin heavy chain (MyHC)-1 (Figure 2A) and anti-MyHC2 (Figure 2B) antibodies. Myotubes treated with 100 µM PA displayed reduced diameter compared to vehicle-treated myotubes irrespective of fiber type; however, they were increased in number (Figure 2C–F,H). Fusion index was not different between control and 100 µM PA-treated cells (Figure 2G).

In PA-treated cells, we observed higher staining for MyHC-1 than MyHC2 (Figure 2H), suggesting a more oxidative metabolism. Therefore, we evaluated mRNA levels of proteins involved in glycolysis (*Hk*, Hexokinase), fatty acid oxidation (*Acad11*, Acyl-CoA dehydrogenase family member 11), and/or mitochondrial markers (*Pdk4*, Pyruvate dehydrogenase kinase, isozyme 4; *Ppargc1a*, Peroxisome proliferative activated receptor gamma coactivator 1 alpha; *Cpt1b*, Carnitine palmitoyltransferase 1b). Treatment with 100 µM PA for 48 h significantly reduced *Pdk4* expression but not *Ppargc1a* (*p* = 0.08), *Cpt1b* (*p* = 0.13), *Hk* (*p* = 0.78), or *Acad11* (*p* = 0.55), suggesting C2C12 myotubes did not have a higher oxidative capacity after PA treatment (Figure 2I).

In order to understand whether the PA-reduced diameter of C2C12 myotubes is associated with an altered balance between protein synthesis and degradation, we evaluated protein synthesis (puromycin incorporation, Figure 3A) and components of protein degradation (protein levels of atrogin-1 and muscle RING finger 1 (muRF1), Figure 3B). Although PA reduced C2C12 myotube diameter, there was no difference in the expression of proteins involved in this balance (Figure 3A,B).

Altogether, data on treatment of differentiated C2C12 cells with PA for 48 h seem to indicate atrophy, but they are not related to protein synthesis and degradation imbalance.

### 2.3. PA Increases miR-206 Expression in C2C12 Myotubes

Considering the importance of microRNAs in the regulation of skeletal muscle phenotype, we measured the expression of miR-1, -133a, -206, -208b, and -499 (microRNAs that are involved in the specification of myofibers, myogenesis, and muscle growth), and miR-23a (microRNA associated with atrophy) in PA-treated and control myotubes (Figure 4). Our results showed that only miR-206 presented a significant increase in the 100 µM PA group compared with the control group (Figure 4). It should be noted that miR-206 is associated with a slow-twitch muscle phenotype [13] and PA induced an increase in the proportion of type I fibers.

### 2.4. PA Alters Expression of Myogenic Markers during Myoblast Differentiation

Muscle mass maintenance is a dynamic process, and the skeletal muscle responds to different stimuli reshaping the morphological, biochemical, and physiological state of the myofibers. Muscle regrowth after injury is crucial for the repair of muscular functions, and in order to elucidate if PA affects C2C12 myogenesis, C2C12 were grown for 5 days in differentiation media in the absence (vehicle-control) or in the presence of 100 or 150 μM PA. For those experiments, we included 150 μM PA concentration, because in the initial experiments, this concentration affected C2C12 myoblasts proliferation.

First, we analyzed the temporal expression of some genes involved in skeletal muscle cell proliferation and differentiation (e.g., myoblast determination protein 1 (*MyoD*), Myogenic factor 5 (*Myf5*), Myomesin-2 (*Myom2*), and Myosin heavy chain 7 (*MyH7*)) and myostatin (*Mtsn*), which is a negative growth regulating factor responsible for inhibition of the PI3K/AKT pathway that acts on protein synthesis [14,15]. Considering *MyoD* and *Myf5*, two early differentiation markers of myotubes, PA has a transient impact on their expression. On day one of differentiation, 100 μM PA increases *Myf5* expression 4-fold, but it decreases its expression from day two to five. On day two of differentiation, 150 μM PA increases *MyoD* expression 6-fold but decreases it on the third day (Figure 5A). We also evaluated the *Myh7* and *Myom2* genes, which are two late differentiation markers of myotubes. While in control cells, a peak of *Myh7* gene expression is observed on day three of differentiation, in PA-treated cells, its expression is significantly reduced. *Myom2* expression is increased by PA treatment compared to vehicle control cells, independently of time (Figure 5A). Considering only time effect, *Myom2* levels decrease on the fifth day of differentiation. Moreover, we observed an increment in *Mstn* levels in the control group on the fifth day of differentiation, but this is not evident in cells treated with PA (Figure 5A).

Altogether, these data suggest that PA dysregulates myogenic markers expression during differentiation and may result in the impaired fusion and growth of myotubes, which can affect muscle function.

### 2.5. PA Affects Temporal Expression of miR-133a and miR-206 during Myoblast Differentiation

Next, we analyzed the three main myomiRs—miR-1, miR-206, and miR-133—involved in myogenesis during each day of differentiation in order to evaluate their temporal profile.

Our results show that the miR-1 pattern of expression in PA-treated cells is not different from control; however, there was an effect of time (*p* = 0.0002) independent of the treatment (Figure 5B). Considering miR-206, in control cells, there is a sustained increase in miR-206 expression between day two and five. In PA-treated cells, treatment with 100 µM reduces miR-206 expression on the fifth day compared to the third day, and treatment with 150 µM decreases miR-206 levels on the third day of differentiation compared to control and 100 µM PA groups (Figure 5B).

In regard to miR-133a levels, treatment of C2C12 cells with 150 µM PA transiently increases its expression on the second day of differentiation compared to control and 100 µM PA and promptly reduces it from the third day on (Figure 5B).

Based on these data, we can suggest that 100 µM PA concentrations affect only miR-206, which is a pro-differentiation and anti-proliferative miRNA, while 150 µM PA also decreases miR-133a, which is a more pro-proliferative microRNA. This latter effect is consistent with the reduced proliferation induced by higher concentration of PA.

### 2.6. PA Impairs the Growth and Promotes Metabolic Remodeling of C2C12 Myotubes

In order to understand if an altered expression of myoblast differentiation markers and miRNAs resulted in smaller myotubes, myoblasts were differentiated and treated with PA for five days. Next, the cells were immunolabeled with anti-MyHC-1 or anti-MyHC-2, which are late markers of differentiated myotubes. For both type I and type II fibers, we could observe that in the presence of 100 µM or 150 µM PA, myoblasts differentiated into myotubes with reduced diameter when compared to control myoblasts (Figure 6A–C). Then, a fusion index was calculated, and PA-treated cells had a significantly lower fusion index compared to control cells (control: 44% versus 100 µM: 17% or 150 µM: 15%) (Figure 6D). Interestingly, for 100 μM PA concentration, there was a reduction of fibers stained for MyHC-1, and for both PA concentrations, there was an increase in fibers stained for MyHC-2, suggesting a switch from slow to fast phenotype in C2C12 myotubes treated with PA (Figure 6E).

Confirming a shift to a more glycolytic phenotype, we observed a robust induction of *HK* transcript in C2C12 myotubes treated with 100 and 150 µM PA (Figure 6F). *Pdk4*, *Ppargc1a*, and *Acad11* mRNA levels were also reduced in 100 µM PA-treated cells (Figure 6F), which is in agreement with the reduced number of type I fibers observed in this condition.

Next, we measured the mRNA expression of genes involved in protein synthesis. In 100 µM PA-treated myotubes, the expression of insulin-like growth factor-1 (*Igf 1*) and *S6k* mRNA was reduced compared to the control group (Figure 6G). There was no statistical difference in Igf1 receptor (*Igf1r*) and Ras homolog enriched in brain (*Rheb*) transcripts levels among the groups, and only 150 µM PA induced *Mtor* levels (Figure 6G). Global protein synthesis measured by puromycin incorporation confirmed that only 100 µM PA decreases myotubes protein synthesis (Figure 6H). Atrogin levels, a marker of protein breakdown, were not different among the groups (Figure 6I). Since Atrogin-1 is inversely related to MyoD expression during the course of C2C12 differentiation [16], we evaluated Atrogin-1 on the first day of PA treatment and found that Atrogin-1 levels were increased by PA treatment in a dose-dependent manner (Figure 6I).

## 3. Discussion

Our research provides new mechanism insights on the effect of chronic PA exposure on skeletal muscle and may help explain why regenerative capacity and muscle regrowth is also impaired in a diet-induced obesity animal model. We found that PA induces atrophy and increases the number of slow fibers, which was correlated with increased miR-206 levels. Then, using myoblasts, we show that PA impairs myotube formation by different mechanisms that seem to be dose-dependent. Here, 150 µM PA reduces cell proliferation after injury and decreases the expression of *MyoD*, miR-206, and miR-133a and increases protein degradation without increasing protein synthesis in regrowing muscle. On the other hand, 100 µM PA did not affect cell proliferation, but it altered *Myf5* expression during differentiation, possibly reducing the fusion of myotubes. Moreover, at this concentration, PA decreases protein synthesis in myotubes via the Igf-1 pathway and downregulates miR-206 expression. At both concentrations, the proportion of fast glycolytic fibers is increased in PA-treated myotubes, which may imply a reduced oxidative capacity.

Previous work on the effect of PA in C2C12 cells has suggested that PA induces atrophy in C2C12 myotubes [7,8,9]. However, the dose of 300 or 500 µM PA was used and, as recently shown by [17] and in this study, a 300 and 500 µM PA dose are both cytotoxic for myoblasts and induce a loss of cell viability. Herein, we chose two non-cytotoxic concentrations to better understand how excess PA could impact myotube formation or diameter without inducing cell death.

The toxicity of PA for cells depends on the level of their unbound fraction rather than on their total concentration [18]. Under conditions observed in human physiology, the most abundant free fatty acids, including PA, in equilibrium with albumin is less than 15 nM [19]. More importantly, in obesity, a 20 nM of unbound palmitate in serum is expected [20]. The estimated unbound palmitate after the addition of 500 µM in the presence of 1% albumin is 27 nM [18], which is almost double that of those found in circulation. Thus, studies of PA-induced atrophy that used 300–500 µM of PA seem unrealistic, as the delivery of a higher amount (27 nM) of unbound PA than would be expected in obesity (20 nM) causes cell death rather than cell atrophy.

In our study with myotubes, miR-206 seems to be a target of PA to influence cell size and myofiber specification. Corroborating with our results, the specific inhibition of miR-206 has been shown to increase C2C12 myotubes width, resulting in hypertrophy in vitro; this effect was not observed when miR-133a was inhibited or overexpressed [21]. In addition, miR-206 is necessary to enforce a slow skeletal muscle phenotype, and miR-206 knockout increases the myofiber cross-sectional area [13]. In spite of the fact that PA-treated myotubes had increased numbers of slow fibers, measurements of transcripts involved in mitochondrial oxidation were not increased. This is in line with previous observation that the exposure of C2C12 myotubes to PA acutely increases mitochondrial respiration, which could reflect increased fatty acid oxidation, but chronic PA treatment causes a marked impairment in mitochondrial function [22].

Deficits in muscle repair are present in diet-induced obese mice, and the impairments in the functionality of muscle satellite cells population, as a result of altered hepatocyte growth factor/c-Met signaling, are contributors to the delayed regeneration [23]. Additionally, obese mice have a general lack of regenerative response at all time points compared to their normal weight counterparts, thus further illustrating the negative influence of obesity on the healing process after skeletal muscle injury [9]. Corroborating these reports, our results suggest that 150 µM PA reduces myoblast proliferation capacity after a scratch wound-healing assay.

It has been demonstrated that the inhibition of any myogenic factors is capable of preventing or decreasing the fusion process, damaging the formation of myotubes [24]. The silencing of *MyF5* gene expression induces a decrease in the replication of myoblasts and a reduction in the number, diameter, and length of myotubes [25]. In our study, 100 µM PA-treated cells had a lower fusion index than the control, corroborating with a reduced expression of myogenic factors and inhibition of growth.

In developing C2C12 cells, there is strong support for a direct link between MyoD and Atrogin-1, as their expression was inversely related during the course of C2C12 differentiation. The overexpression of Atrogin suppresses MyoD-induced differentiation, inhibiting myotube formation [16]. Furthermore, during myogenic differentiation, an up-regulation of miR-133a is required for myoblast–myotube transition, which is induced by MyoD [26,27]. Accordingly, in this study, we observed in 150 µM PA-treated cells an up-regulation of Atrogin-1 followed by the reduced expression of MyoD and miR-133a.

miR-206 expression is detectable in C2C12 myoblasts, but its steady-state level increases many fold as C2C12 cells differentiate [27]. In our study, PA significantly reduced miR-206 expression and as mentioned earlier, miR-206 can specifically regulate the growth of myotubes.

In animals, deletion of the *Mstn* gene leads to a significant increase in muscle mass; at the same time, the stimulation of *Mstn* yielded a decrease in mass, suggesting that myostatin is a negative regulator of muscle development [15,28]. Interestingly, *Igf-1* and *Igf-2* are necessary for myostatin to induce hypertrophy [29]. We noted that after 5 days of differentiation, *Mstn* and *Igf-1* were reduced in 100 µM PA-treated cells.

A crosstalk between insulin and Igf-1 is mediated by the phosphorylation of Irs1 residues [29,30]; thus, insulin/IGF-1 signaling could underlie the poor healing of injured muscles that is associated with diabetes [31]. Supporting these hypotheses, it has been demonstrated that PA in cultured muscle cells increases the activity of phosphatase and tensin homolog (PTEN) (which antagonizes the P13K signaling pathway), decreasing PIP3 and leading to the decreased phosphorylation of AKT and S6K and suppressed growth [31,32].

A lower proportion of type I fibers in skeletal muscle of individuals with metabolic syndrome has been described, and the lower proportion of these fibers correlates with the severity of insulin resistance [33]. In addition, both obese and type 2 diabetics individuals present a lower proportion of type I fibers and an increase in fibers type IIX in skeletal muscle [34]. This is in agreement with our data describing a more glycolytic phenotype in PA-treated myotubes.

Previously, it was shown that the treatment of C2C12 myotubes with PA for 24 h decreases oxidative capacity [22]. Thus, PA induction of fast type fibers may imply a reduced oxidative capacity and allow for greater susceptibility to injured muscle cells, as type II fibers are more susceptible to damage than type I fibers [35].

The mechanism by which PA increases the proportion of fast fibers may be related to the induction of *Myom2* and reduction of *Myh7* levels during differentiation. The *Myh7* gene is expressed in slow type I fibers that exhibit oxidative metabolism [36]; on the other hand, *Myom2* is initially present in both slow- and fast-skeletal muscle embryonic fibers, and it is typically suppressed in slow fibers but present in fast type fibers [37,38].

Another postulated mechanism is the down-regulation of miR-206 levels. MiR-206 is enriched in slow twitch muscles [39] and is essential to enforce a slow muscle program in skeletal muscle [13]. It is worth noting that in skeletal muscle of diabetic patients, miR-206 expression is reduced [40], as well as the proportion of fibers type IIX, as pointed out before. Therefore, miR-206 may be associated with slow-to-fast fiber transition in myotubes differentiated in the presence of PA.

In conclusion, herein, we provided data indicating that PA-induced impaired myogenesis and a more glycolytic phenotype may be due to dysregulation of myomiRs. Since PA is enriched in high-fat diets and sarcopenic obesity is an increasing health problem, by providing new insights into the mechanism by which PA negatively impacts skeletal muscle, we may suggest new drug candidates to treat sarcopenic obesity.

## 4. Materials and Methods

### 4.1. Cell Culture and Cell Treatments

Mouse C2C12 myoblasts were purchased from American Type Culture Collection (ATCC, no: CRL-1772) and used until passage 8. Myoblasts were cultured in Dulbecco’s Modified Eagle Medium (DMEM) with low glucose concentration, supplemented with 10% fetal bovine serum (FBS) (Life Technologies, Long Island, NY, USA) and 1% antibiotic (50 IU/mL penicillin, 50 μg/mL streptomycin) and maintained in a humidified incubator at 37 °C in an atmosphere of 5% CO_2_. Cells were seeded in 6-well plates, 100,000 cells per well, and when they reached 100% confluence, they were incubated with DMEM supplemented with 2% horse serum for 5 days to induce myotubes differentiation. PA (Palmitic acid, P5582, Sigma-Aldrich, St. Louis, MO, USA) was dissolved in ethanol 100% and diluted in DMEM supplemented with 10% FBS or with 2% horse serum (Life Technologies) depending on the assay, containing 2% fatty acid-free bovine serum albumin (BSA) (A9205, Sigma-Aldrich) to reach desired fatty acid concentrations [24]. For the experiments, we used undifferentiated cells treated with 100 μM and 150 μM PA containing 2% BSA for 1 to 5 days or differentiated cells for 5 days and treated with PA for the subsequent 48 h (7-day differentiated cells). In both experiments, fresh media was provided every 24 h. A control group containing vehicle (DMEM, 2% BSA, and 0.19% ethanol) was also included in all experiments.

### 4.2. Cytotoxicity Assays

The effect of PA on the cytotoxicity of C2C12 myoblasts was evaluated after 24–72 h of treatment with PA (0–500 µM). For MTT assay, 400 µL of MTT solution (diluted in DMEM) was added to each well of a 6-well plate, and then, the plates were incubated at 37 °C for a duration of 2.5 h. After this period, the medium containing MTT was removed, and 700 µL of DMSO was added to the wells to solubilize the formazan salt. Then, the solutions were homogenized, and 200 µL was transferred to a 96-well plate for spectrophotometry analysis, and absorbance was measured at 550 nm. For lactate dehydrogenase (LDH) assay (Cytotoxicity Detection Kit—Roche Diagnostics GmbH, Mannheim, Germany), at the end of the period of treatment, the medium was collected, and 30 µL was transferred to a 96-well plate containing 70 µL of LDH solution. The plate was incubated at room temperature for 30 min, and then, absorbance at 492 nm was measured.

TUNEL (TdT-mediated dUTP-TMR Nick End Labeling) assay was used to quantify apoptotic cells using In Situ Cell Death Detection Kits (Roche). Myoblasts were grown on 6-well plates until they reached 100% confluence and then were treated with vehicle or PA (100 μM and 150 μM) for 24 h. Cells were fixed in 4% paraformaldehyde and permeabilized with 0.1% Triton X-100 in 0.05 M phosphate-buffered saline (PBS) containing 0.1% sodium citrate. Subsequently, the cells were incubated with the enzyme terminal deoxynucleotidyl transferase (TdT) and deoxyadenosine triphosphate (dATP), labeled with Tetramethyl rhodamine diluted 1:5 (GIBCO) and DAPI (1:10,000), at 37 °C for 1 h. The cells were washed with phosphate buffer saline and visualized with microscope fluorescence (Carl Zeiss) using AxioVision 4.8 software (Carl Zeiss, Jena, Germany).

### 4.3. Healing Assay

Cells were seeded in 6-well plates and grown to 100% confluence. The cell layer was scratched using a sterile pipette tip (p200), and the distance of the resulting gap was measured initially (time 0 h) and after 2, 12, and 16 h of the injury. Cells were photographed using a camera (Axiocam MRC Zeiss, Carl Zeiss, Jena, Germany) coupled to an inverted light microscope (Zeiss Observer A1, Carl Zeiss, Jena, Germany). Translocation (net distance by a cell during the course of the experiment) and speed of migration were evaluated. The distance of the gaps was measured using the AxioVision 4.8 software.

### 4.4. Immunofluorescence

Myoblasts were cultured and treated with PA during 5 days of differentiation into myotubes, or 5-day myotubes were treated for 48 h with PA. In both conditions, cells were washed 1x with PBS and fixed with methanol for 20 min. Subsequently, cells were washed 3x with PBS and incubated with 5% blocking solution (1× PBS + 5% serum + 0.05% Triton X-100) for 2 h at room temperature. After washing 3x with PBS, cells were incubated with anti-MyHc1 (for type I, slow fiber M8421 Sigma-Aldrich) or anti-MyHC2 (for type II, fast fiber, M4276 Sigma-Aldrich) overnight at 4 °C (1:250), washed with PBS (3 times for 10 min), and incubated with fluorescein isothiocyanate (FITC)-conjugated goat anti-mouse IgG (1:500 dilution Sigma-Aldrich) and DAPI for 2 h at 4 °C, washed with PBS, and then examined immediately in fluorescence microscope (Carl Zeiss). Images from positive myotubes for anti-MyHC1 and anti-MyHC2 were captured at 200X magnification using an Axiocam MRC Zeiss digital camera. For each well of a 6-well plate, four regions were used, and at least thirty myotubes were analyzed per well and quantified using AxioVision 4.8 software. Both cell diameter and length were measured.

### 4.5. RNA Preparation and RT-qPCR

Total RNA from C2C12 cells grown for 1–5 days in the absence or presence of PA and from differentiated myotubes treated or not with PA for 48 h was extracted using TRIzol^®^ reagent (Life Technologies, Carlsbad, CA, USA), according to the manufacturer’s instructions. Isolated RNA was quantified using a BioDrop µLite spectrophotometer (BioDrop, Cambridge, England, UK). RNA was stored at −80 °C prior to analysis by RT-qPCR. For miRNA qPCR, the RNA samples were quantified as previously described, and 50ng/reaction of total RNA were transcribed into cDNA using a Taqman^®^ microRNA assay system (Applied Biosystems, Foster City, CA, USA), according to the manufacturer’s instructions. The expression of miRNA miR-1, miR-206, miR-133a, miR-208b, miR- 23a, and miR-499 was measured by qPCR, using ABI Prism 7500 equipment (Applied Biosystems, Foster City, CA, USA), following the universal protocol of amplification: 95 °C for 10 min, 40 cycles of 95 °C for 15 s, and 60 °C for 1 min, followed by dissociation curve. Gene expression was quantified using the 2−ΔΔCT method. The housekeeping genes *U6* and *sno202* were used as normalizers for myoblasts and myotubes, respectively. For mRNA expression, cDNA was synthesized from 500 ng total RNA extract using SuperScriptTM III RT RNase H (Life Technologies), according to manufacturer’s instructions. The quantification of mRNA expression of genes involved in metabolism was performed by RT-qPCR in a total volume of 10 μL, containing diluted cDNA template (1/10), forward and reverse primers (200 nM each), and SYBR Green Master Mix (Life Technologies). Gene expression was performed using the 7500 Real Time PCR System (Applied Biosystems), following the universal protocol of amplification: 95 °C for 10 min, 40 cycles of 95 °C for 15 s, and 60 °C for 1 min. Dissociation curves were performed to test primers specificity. Relative gene expression quantification was determined by 2−ΔΔCT method. *Hprt1 or Rpl0* was used as a reference gene for myoblast and myotubes, respectively. Primers sequences are shown in Table 1.

### 4.6. Western Blotting

Proteins from 7-day differentiated myotubes treated with PA were extracted by homogenizing the myotubes in radioimmunoprecipitation assay (RIPA) iced buffer containing 1% protease inhibitor cocktail, and concentration was determined by Bradford method. Samples containing 50 μg of proteins in the sample buffer were separated by electrophoresis using 8% or 10% SDS-PAGE gels and transferred to a nitrocellulose membrane (250 mA for 2 h). Membranes were blocked in 5% non-fat milk diluted in TBS-T buffer for 1 h at room temperature and incubated overnight at 4 °C with the following primary antibodies: MurF1 (1:1000, MP3401, ECM Biosciences, Versailles, KY, USA), atrogin-1 (1:1000, AP2041, ECM Biosciences), and GAPDH (1:5000, ab181602, Abcam). Detection was performed by a C-Digit Imager (LI-COR, Lincoln, NE) after incubation with peroxidase-labeled secondary antibody (1:10,000) for 2 h at room temperature, using a Pierce ECL Western blotting substrate detection system (Thermo Scientific). Signals were quantified using NIH ImageJ 1.63 Software using GAPDH as a reference.

### 4.7. Puromycin Incorporation Assay

First, 1 µM puromycin diluted in water was added directly to the cells previously treated with PA and incubated for 15 min. Cells were washed twice with ice cold PBS and cells were lysed with sucrose buffer. Cells were centrifuged at 10,000× *g* for 1 min, supernatant was removed, and protein was measured using the Bradford method. The Western blot procedure was used to quantify puromycin incorporation using an anti-puromycin antibody (1:1000 Millipore, MABE343). Signals were normalized to total protein content stained with ponceau.

### 4.8. Statistical Analysis

The results are expressed as mean ± standard error of the mean (SEM) and analyzed by an unpaired two-tailed *t*-test, one-way or two-way ANOVA followed by Tukey’s post-test. For two-way ANOVA, when there was no interaction between the two factors analyzed (time and PA treatment), the factors were considered separately, and when interaction was statistically significant, Tukey’s post-test was performed. Statistical analyses were performed using GraphPad Prism version 8.0 for Windows (GraphPad Software, San Diego, CA, USA).

## Figures and Tables

**Figure 1 ijms-22-02748-f001:**
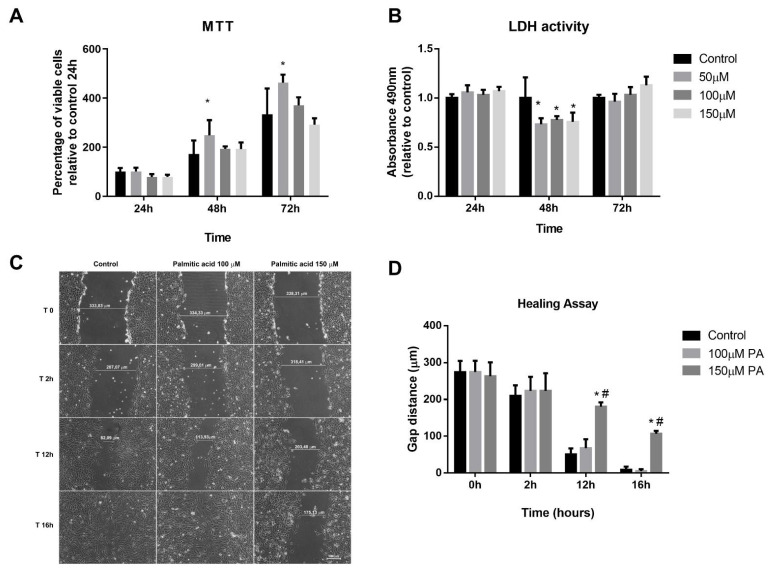
Palmitic acid (PA) does not cause cytotoxicity but affects proliferation in a dose-dependent manner. (**A**) MTT (3-(4,5-dimethyl-2-thiazolyl)-2,5-diphenyl-2H-tetrazolium bromide) and (**B**) lactate dehydrogenase (LDH) assays in myoblasts at 24, 48, and 72 h; * *p* < 0.05 evaluated by two-way ANOVA, followed by Tukey post-test. * versus control. (**C**,**D**) Representative images of myoblast proliferation during healing assay and quantification of the gap distances along the time. *^, #^
*p* < 0.05 evaluated by two-way ANOVA repeated measures, followed by Tukey post-test. * versus control and ^#^ versus 100 µM. For all experiments, *n* = 3.

**Figure 2 ijms-22-02748-f002:**
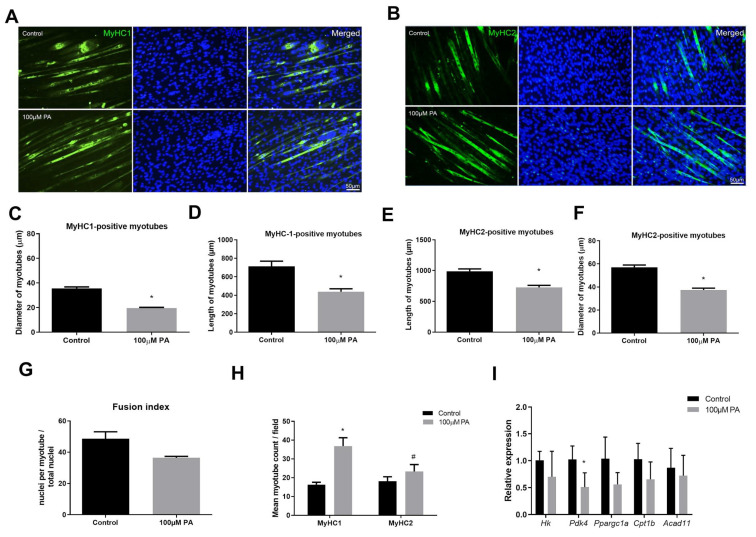
Palmitic acid treatment of C2C12 myoblasts results in smaller myotubes. Type I (**A**) and Type II (**B**) (green) myotubes were identified by immunofluorescence in differentiated C2C12 myotubes treated with vehicle (Control) or 100 μM PA for 48 h. Nuclei were stained with diamidino-2-phenylindole (DAPI) (blue). Length and diameter of type I (**C**,**D**) and type II (**E**,**F**) were measured; (**G**) number of stained myotubes for anti-MyHC1 or anti-MyHC2 were counted per field; (**H**) Fusion index; (**I**) mRNA expression of genes related to glycolysis, fatty acid oxidation, and mitochondria were measured by RT-qPCR. mRNA expression was normalized to ribosomal protein, large P0 (*Rpl0*) expression and are expressed as relative to control levels *^,#^
*p* < 0.05 as indicated by two-way ANOVA followed by Tukey’s test (* versus Control; ^#^ versus MyHC1 100 μM PA); *n* = 5.

**Figure 3 ijms-22-02748-f003:**
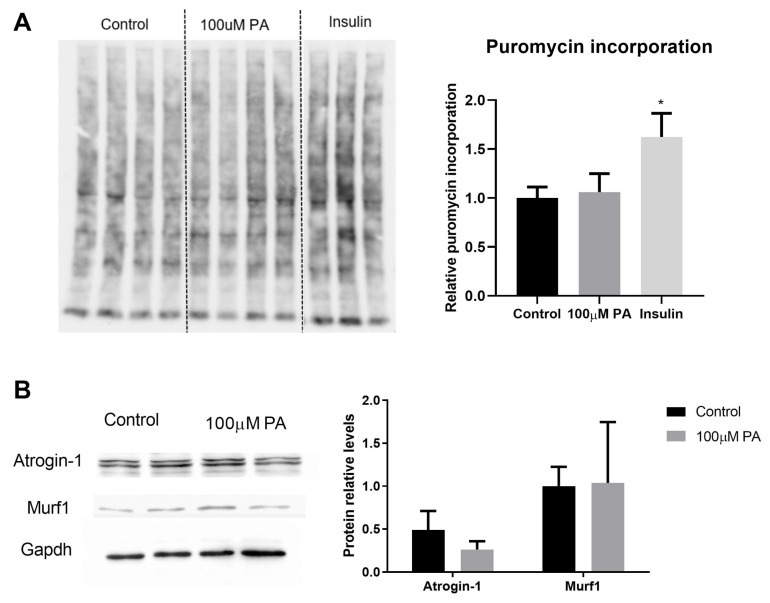
Effect of palmitic acid on protein synthesis and degradation. (**A**) Puromycin incorporation assay (*n* = 4). Cells treated with insulin for 30 min were used as a positive control of increased protein synthesis. Puromycin incorporation was normalized to total protein content detected by Ponceau staining, * *p* < 0.05; (**B**) Expression of proteins involved in protein degradation (Atrogin-1 and muRF1) was measured by Western blot in differentiated C2C12 myotubes treated for 48 h with 100 µM PA and normalized to GAPDH levels (*n* = 4). T test was performed, *p* > 0.05.

**Figure 4 ijms-22-02748-f004:**
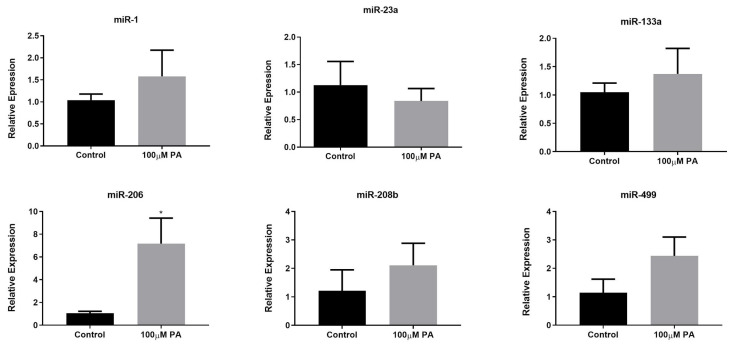
MicroRNA expression in differentiated myotubes treated with vehicle (Control) or 100 µM PA for 48 h. miRNAs expression was measured by Stem-loop RT-qPCR. * *p* < 0.05 versus Control, indicated by *t* test; *n* = 5.

**Figure 5 ijms-22-02748-f005:**
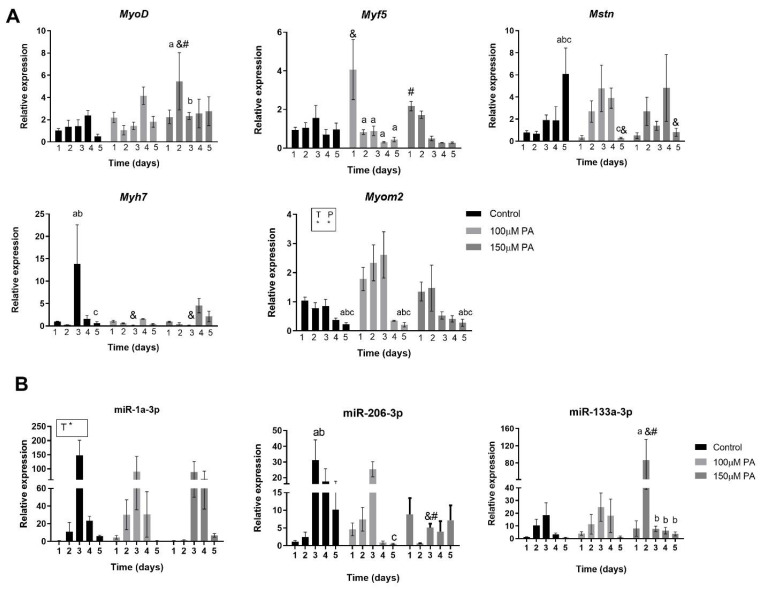
PA alters the temporal expression of myogenic marker and microRNAs. Myogenic markers mRNA expression (**A**) and expression of miR-1, miR-133a, and miR-206 (**B**). C2C12 were grown for 5 days in differentiation media in the absence (vehicle-control) or in the presence of 100 and 150 μM PA (*n* = 3). Fold change in the expression of target mRNA and miRNA are relative to *Rpl0* and *U6*, respectively, which are used as internal control genes. * *p* < 0.05, two-way ANOVA for time (T) or P (palmitic acid). ^&,#,a,b,c^
*p* < 0.05, as indicated by Tukey’s post-test, only performed when interaction was significant. ^&^ versus Control; ^#^ versus 100 μM PA; ^abc^ versus day 1, 2, 3, and 4, respectively within each concentration.

**Figure 6 ijms-22-02748-f006:**
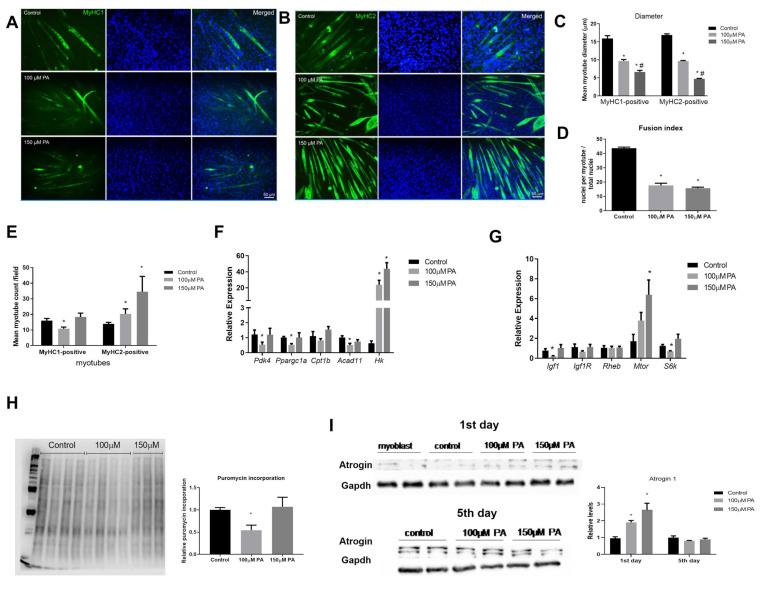
PA impairs myotube formation and induces a slow-to-fast fiber transition in C2C12 myotubes. Immunofluorescence analysis of MyHC1 (**A**) and MyHC2 (**B**) (green) proteins. C2C12 were grown for five days in differentiation media in the absence (vehicle-control) or in the presence of 100 or 150 μM PA. Nuclei were stained with DAPI (blue). Diameters of type I and type II were measured (**C**). Fusion index (**D**). MyHc1 and MyHC2-positive myotubes counted per field (at least four fields were counted per well) (**E**). Relative expression of metabolic genes by RT-qPCR. mRNA expression was normalized to *Rpl0* expression (**F**). Relative expression of genes involved in protein synthesis. Gene expression was normalized to *Rpl0* levels and is expressed relative to control (**G**). Representative blots of puromycin incorporation (**H**) and Atrogin-1 (**I**) Western blot. Densitometry of bands were analyzed and normalized by Ponceau staining (*n* = 6) (**H**) and GAPDH content (*n* = 4) (**I**). *^,#^
*p* < 0.05, evaluated by one-way ANOVA followed by Tukey’s post-test * *versus* Control and ^#^
*versus* 100 μM PA. *n* = 5.

**Table 1 ijms-22-02748-t001:** Primers sequences used to quantify mRNA expression by qPCR.

Gene Symbol	Forward Primer Sequence	Reverse Primer Sequence
*Pdk4*	GTTCCTTCACACCTTCACCACA	CCTCCTCGGTCAGAAATCTTGA
*Ppargc1a*	CACCAAACCCACAGAAAACAG	GGGTCAGAGGAAGAGATAAAGTTG
*Cpt1b*	CCTCCGAAAAGCACCAAAAC	GCTCCAGGGTTCAGAAAGTAC
*Igf1*	GTGAGCCAAAGACACACCCA	ACCTCTGATTTTCCGAGTTGC
*Igf1R*	CTCTGTTACCTCTCCACCAT	CTTCTCACACATGGGCTTCT
*S6K*	GGAGGGACAGAAGAGAATCA	AACCTAGAACCACACCAATG
*Rheb*	CGATCCAACCATAGAGAACAC	AATATTCATCCTGCCCCGCT
*Mtor*	TGCCGCTGAGAGATGACAATG	GTTGTTAATGCTGATGAGGG
*Mstn*	GCAAAATTGGCTCAAACAGCC	AGGGATTCAGCCCATCTTCTC
*Myf5*	TGACGGCATGCCTGAATGT	GCTGGACAAGCAATCCAAGC
*MyoD*	GCCCGCGCTCCAACTGCTCTGAT	CCTACGGTGGTGCGCCCTCTGC
*MyH7*	GCCAACTATGCTGGAGCTGATGCCC	GGTGCGTGGAGCGCAAGTTTGTCATAAG
*Myom 2*	GATCAACAGGGCCAACTTTGA	TGGTAGACACTTGTTCATGGGAAT
*Acad11*	AGATGCTTCAGTTATCGGAACG	ATGTAGCCATGCCAGGGTTTC
*Hk*	GCTCAGAAAAGGGGGATTTC	TCAGCGACGTGATCAAAAAG
*Rpl0*	TAAAGACTGGAGACAAGGTG	GTGTACTCAGTCTCCACAGA

## Data Availability

The data presented in this study are available on request from the corresponding author.

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
