# Peer review of "Palmitic Acid Impairs Myogenesis and Alters Temporal Expression of miR-133a and miR-206 in C2C12 Myoblasts"

_ijms, 2021, doi:10.3390/ijms22052748_

Round 1

Reviewer 1 Report

The manuscript titled Palmitic acid impairs myogenesis and alters temporal expression of miR-133a and miR-206 in C2C12 myoblasts by da Paixão et al attempted to identify effects of relatively lower concentrations of PA on in vitro C2C12 viability, proliferative capacity, myogenesis and myotube size. These parameters were also investigated in relation to gene protein and miRNA expression. The use of PA in in-vitro models of muscle regeneration has been explored thoroughly by many investigators the current study provides additional information of miRNA adding some "new" contribution to the field. Comments for consideration 1. English editing is required for example L41-44 a long and structurally incorrect sentence L55-L59 a long and incorrect sentence with missing some connective/stop words 2. L41 what the authors mean by muscle maintenance? 3. L75,76 “We first attempted to use 250μM and 500μM concentrations; 75 however, these concentrations significantly reduced cell viability after 24h (87% and 54%, 76 respectively)”. Is the lower concentration induced higher inhibition and higher cell death, if yeas, how the authors explain this 4.Discussion is including a repetition of results for example L273-278 needs to refrain from doing so. Additionally, the discussion is missing a conclusive section. The conclusion section L472 is negligible and should be expended.

Reviewer 2 Report

The authors performed a study on the effect of palmitic acid (PA) on the proliferation-differentiation of C2C12 cells in myotubes. To reach their conclusions, the authors studied molecular markers associated with myotube differentiation, using non-cytotoxic concentrations of PA. Immunofluorescence techniques were performed to evaluate the number and distribution and growth parameters of the MyHC1 and MyHC2 muscle fiber types.

The authors conclude that PA treatment affects different steps in myotube formation such as proliferation, differentiation, and fusion, resulting in smaller and more glycolytic myofibers in C2C12 cells.

The work presents a series of data that do not always support the authors' claims. Often the results obtained with the two concentrations used (100 uM and 150 uM) of PA are discordant, so it becomes impossible to reach the conclusions indicated by the authors. In some key experiments of the work, it is essential to carry out control experiments (silencing of myogenic factors, or miRNAs) in order to strengthen the conclusions.

The text presents numerous grammatical errors, and a thorough review of the manuscript by a native speaker is strongly recommended. The figures show a series of imperfections (for example, the standard deviation bars in Figure 2I, the position of gene names on the abscissa, in Figure 2I, the position of the asterisks in Figure 6, etc).

Below are the points that the authors can review to improve the manuscript:

Line 62, the authors indicate that there are no differences in the expression of proteins involved in protein synthesis. This claim is not supported by experimental evidence.

Lines 79-81, it is not clear why the authors performed the TUNEL test since no alterations in viability were detected. These images are superfluous and can go into supplementary data.

Lines 88-90, The authors performed the cytotoxicity test on myoblasts (undifferentiated cells), then they claim to use the 100 uM PA concentration to avoid the death of the differentiated myotubes. From a methodological point of view, this procedure is not correct. Furthermore, it is not established that, once differentiated, myotubes can proliferate and whether PA can influence myotube proliferation.

Lines 106-107, it is not clear what the authors intend to claim.

Lines 107-108. The authors should explain why they analyzed IRS1, PDK4, PGC1a, and CPT1 to assess glucose and fatty acid metabolism. Why didn't the authors analyzed the expression of glycolytic genes or genes involved in the beta-oxidation of fatty acids?

Lines 125-132. The results on the mechanisms regulating the overall protein content in cells are inconsistent with the observations about the reduced dimension (diameter and length) of myotubes. In my opinion, the discrepancies may be associated with too few differentiated myotubes versus more undifferentiated cells. For this reason, the authors' conclusions cannot be validated following the methodology followed. Authors should use conditions to obtain a greater number of differentiated cells. Alternatively, the authors can isolate differentiated cells from undifferentiated ones. In addition, it would have been useful to use a positive control (treatment with insulin) in immunofluorescence analyzes.

Line 150-152: miRNAs are involved in the translation of mRNA, but not in epigenesis. The statement is incorrect.

Lines 233-234, why did the authors use a different method to assess reduced protein synthesis? Has it been shown in other studies whether the reduction in mRNA levels for Igf1 and S6k in C2C12 is correlated with reduced protein synthesis? It is not understandable why the same observations obtained with 100 uM PA are not observed with 150 uM PA.

Lines 243-243, there are conflicting data regarding the effect of PA on the reduction of slow oxidative type I fiber

Lines 269-271, Contrary to what the authors stated, the results obtained do not seem to be supported by previous studies (21). In fact, according to 21, PA increases the expression of UCP3 which is essential to support the beta-oxidation of fatty acids and to reduce the toxic effect of PA. Conversely, the authors stated that PA at 100 uM reduces the beta-oxidation of fatty acids.

Line 273, what do the authors mean that “150uM PA reduces the proliferation of myoblasts after healing”?

Line 278, What does “MRFs” mean?

Lines 316-317, the authors state that the reduction of IGF1 and S6K to 100uM PA could explain the reduced growth of myotubes. However, this statement is not supported by the data obtained using PA 150 uM.

Line 380 The authors stated that TUNEL test (TdT-mediated dUTP-TMR Nick End Labeling) has been carried out by using dATP labeled with TAMRA. Please indicate the commercial source.

Figure 6, the quantification of IGFR and RHEB is shown, but they were not reported and commented on the manuscript.

Round 2

Reviewer 2 Report

The authors presented a revised version of the manuscript "Palmitic acid impairs myogenesis and alters temporal expression of miR-133a and miR-206 in C2C12 myoblasts", taking into consideration the observations I made in the first revision. In this version, the authors added some new experimental data, rectified the imperfections of the figures, reworked the discussion, and introduced supporting elements explaining some inconsistencies that emerged in the first revision phase. Therefore, I can consider the new version suitable for publication in IJMS.

Author Response

The authors thank the reviewer for carefully reading the manuscript and for all suggestions. We reviewed English language are the correction made are highlighted in red.

Please see lines 61-65; 86; 133-114; 127-128; 175; 177-178; 203; 205; 224-225 ; 296; 313-315; 317